# Equality of Opportunity in Classification: A Causal Approach

**Junzhe Zhang**
Purdue University, USA
zhang745@purdue.edu

**Elias Bareinboim**
Purdue University, USA
eb@purdue.edu

## Abstract

The *Equalized Odds* (for short, EO) is one of the most popular measures of discrimination used in the supervised learning setting. It ascertains fairness through the balance of the misclassification rates (false positive and negative) across the protected groups – e.g., in the context of law enforcement, an African-American defendant who would not commit a future crime will have an *equal opportunity* of being released, compared to a non-recidivating Caucasian defendant. Despite this noble goal, it has been acknowledged in the literature that statistical tests based on the EO are oblivious to the underlying causal mechanisms that generated the disparity in the first place (Hardt et al. 2016). This leads to a critical disconnect between statistical measures readable from the data and the meaning of discrimination in the legal system, where compelling evidence that the observed disparity is tied to a specific causal process deemed unfair by society is required to characterize discrimination. The goal of this paper is to develop a principled approach to connect the statistical disparities characterized by the EO and the underlying, elusive, and frequently unobserved, causal mechanisms that generated such inequality. We start by introducing a new family of counterfactual measures that allows one to explain the misclassification disparities in terms of the underlying mechanisms in an arbitrary, non-parametric structural causal model. This will, in turn, allow legal and data analysts to interpret currently deployed classifiers through causal lens, linking the statistical disparities found in the data to the corresponding causal processes. Leveraging the new family of counterfactual measures, we develop a learning procedure to construct a classifier that is statistically efficient, interpretable, and compatible with the basic human intuition of fairness. We demonstrate our results through experiments in both real (COMPAS) and synthetic datasets.

## 1   Introduction

The goal of supervised learning is to provide a statistical basis upon which individuals with different group memberships can be reliably classified. For instance, a bank may want to learn a function from a set of background factors so as to determine whether a customer will repay her loan; a university may train a classifier to predict the future GPA of an applicant to decide whether to accept her into the program. The growing adoption of automated systems based on standard classification algorithms throughout society (including in law enforcement, education, and finance [13, 4, 8, 21, 1]) has raised concerns about potential issues due to unfairness and discrimination.

A recent high-profile example is a risk assessment tool called COMPAS, which has been widely used across the US to inform decisions in the criminal justice system. Fig. 1 graphically describes this setting – $X$ represents the race (0 for Caucasian, 1 for African-American) of a defendant and $Y$ stands

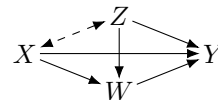

Figure 1: COMPAS

for the recidivism outcome (0 for no, 1 otherwise), which are *mediated* by the prior convictions $W$, and *confounded* by other demographic information $Z$ (e.g., age, gender) of the defendant. The COMPAS

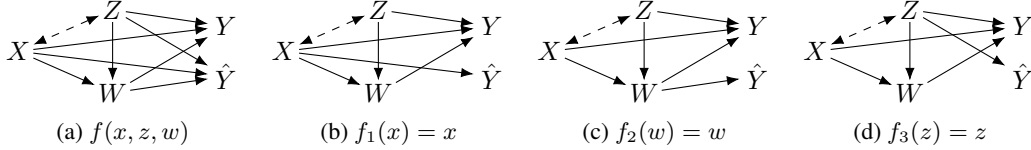

| (a) $f(x, z, w)$ | (b) $f_1(x) = x$ | (c) $f_2(w) = w$ | (d) $f_3(z) = z$ |

Figure 2: (a-d) Causal diagrams of classifiers $f, f_1, f_2, f_3$ in COMPAS. Nodes represent variables, directed arrows for functional relationships, and bi-directed arrows for unknown associations.

tool is a classifier $f(x, z, w)$ (shown in Fig. 2(a)) providing a prediction $\hat{Y}$ on whether the defendant is expected to commit a future crime. An analysis performed by the news organization ProPublica revealed that the odds of receiving a positive prediction ($\hat{Y} = 1$) for defendants who did not recidivate were on average higher among African-Americans than their Caucasians counterparts [1]. In words, the error rates of COMPAS disproportionately misclassified African-American defendants.

Many attempts have been made to model discrimination in the classification setting [26, 14, 11, 9, 15]. A recent, noteworthy framework comes under the rubric of *Equalized Odds* [7] (also referred to as *Error Rate Balance* [5]), which constrains the classification algorithm such that its disparate error rate $ER_{x_0, x_1}(\hat{y}|y) = P(\hat{y}|x_1, y) - P(\hat{y}|x_0, y)$ is *equalized* (and equal to 0) across different demographics $x_0, x_1$, i.e., the odds of misclassification does not disproportionately affect any population sub-group. In the COMPAS example, the condition $ER_{x_0, x_1}(\hat{Y} = 1|Y = 0) = 0$ implies that an African-American defendant who does not commit a future crime will have an *equal opportunity* of getting released, compared to non-recidivating Caucasian defendants. This notion of fairness is natural in many learning settings and, indeed, has been implemented in a number of algorithms [7, 6, 25, 23].

Unfortunately, the framework of equalized odds is not without its problems. To witness, consider a binary instance of Fig. 1 where the values of $X$ and $Z$ are determined such that $x = z$ and $W$ is decided by the function $w \leftarrow x$. We are concerned with the ER disparity induced by different classifiers $f_1, f_2, f_3$ (Fig. 2(b-d)), where, for instance, $\hat{y} \leftarrow f_1(x) = x$ (i.e., $f_1$ takes only $X$ as input, and ignores the other features). Remarkably, a simple analysis shows that $ER_{x_0, x_1}(\hat{Y} = 1|Y = 0)$ is the same (and equal to 1) in all three classifiers, despite their fundamentally different mechanisms associating $X$ and $\hat{Y}$. Note that $f_1, f_2, f_3$ corresponds to the direct path $X \rightarrow \hat{Y}$, the indirect path $X \rightarrow W \rightarrow \hat{Y}$, and the remaining spurious (non-causal) paths (e.g., $X \leftrightarrow Z \rightarrow \hat{Y}$), respectively.

This observation is not entirely new, and is part of a pattern noted by [7] – statistical tests based on the disparate ER are oblivious to the underlying causal mechanisms that generated the data. This realization has dramatic implications to the applicability of supervised learning in the real world since it seems to suggest that commonsense notions of discrimination, for example, the unequalized false positive rate *caused* by direct discrimination ($X \rightarrow \hat{Y}$), cannot be formally articulated, measured from data, and, therefore, controlled. More importantly, the legal frameworks of anti-discrimination laws in the US (e.g., Title VII) require that to establish a *prima facie* case of discrimination, the plaintiff must demonstrate *"a strong causal connection"* between the alleged discriminatory practice and the observed statistical disparity, otherwise the case will be dismissed (Texas Dept. of Housing and Community Affairs v. Inclusive Communities Project, Inc., 576 U.S. __ (2015)). Without a robust causal basis, an evidence of disparate ER on its own is not sufficient to lead to any legal liability.

More recently, the use of causal reasoning to help open the black-box of decision-making systems has attracted considerable interest in the community, leading to fine-grained explanations of observed statistical biases [11, 10, 25, 9]. One of the main tasks of causal inference is to explain "how nature works," or more technically, to decompose a composite statistical measure (e.g, the total variation $TV_{x_0, x_1}(\hat{y}) = P(\hat{y}|x_1) - P(\hat{y}|x_0)$), into its most elementary and interpretable components [24, 17, 29]. In particular, [28] introduced the *causal explanation formula*, which allows fairness analysts to decompose TV into detailed counterfactual measures describing the effects along direct, indirect, and spurious paths from $X$ to $\hat{Y}$. While [28] explains how the statistical inequality in the observed outcome is brought about, it is unclear how to apply such insight to correct the problematic behaviors of an alleged, discriminatory policy. Furthermore, the explanation formula allows the decomposition of marginal measures such as TV, but it's unable to explain disparities represented by conditional ones, such as the ER (e.g., non-recidivating African-American defendants).

This paper aims to overcome these challenges. We develop a causal framework to link the disparities realized through the ER and the (unobserved) causal mechanisms by which the protected attribute $X$

affects change in the prediction $\hat{Y}$. Specifically, (1) we introduce a family of counterfactual measures capable of describing the ER in terms of the direct, indirect, and spurious paths from $X$ to $\hat{Y}$ on an arbitrary structural causal model (Defs. 1-3) and we prove different qualitative and quantitative properties of these measures (Thms. 1-2); (2) we derive adjustment-like formulas to estimate the counterfactual ERs from observational data (Thms. 3-4), which are accompanied with an efficient algorithm (Alg. 1, Thm. 5) to find the corresponding admissible sets; (3) we operationalize the proposed counterfactual estimands through a novel procedure to learn a fair classifier subject to constraints over the effect along the underlying causal mechanisms (Algs. 2-3, Thm. 6).

## 2 Preliminaries and Notations

We use capital letters to denote variables ($X$), and small letters for their values ($x$). We use the abbreviation $P(x)$ to represent the probabilities $P(X = x)$. For arbitrary sets $\boldsymbol{A}$ and $\boldsymbol{B}$, let $\boldsymbol{A}\backslash\boldsymbol{B}$ denote the set difference $\{x : x \in \boldsymbol{A}$ and $x \notin \boldsymbol{B}\}$, and let $|\boldsymbol{A}|$ be the dimension of set $\boldsymbol{A}$.

The basic semantical framework of our analysis rests on *structural causal models* (SCM) [16, Ch. 7]. A SCM is a tuple $\langle M, P(\boldsymbol{u})\rangle$, where $M$ consists of a set of endogenous (observed) variables $\boldsymbol{V}$ and exogenous (unobserved) variables $\boldsymbol{U}$. The values of each $V_i \in \boldsymbol{V}$ are determined by a structural function $f_{V_i}$ taking as arguments a combination of other endogenous and exogenous variables (i.e., $V_i \leftarrow f_{V_i}(PA_i, U_i), PA_i \subseteq \boldsymbol{V}, U_i \subseteq \boldsymbol{U})$). Values of $U$ are drawn from the distribution $P(\boldsymbol{u})$. Each SCM is associated with a directed acyclic graph (DAG) $G = \langle \boldsymbol{V}, \boldsymbol{E}\rangle$, termed a causal diagram, where nodes $\boldsymbol{V}$ represent endogenous variables and directed edges $\boldsymbol{E}$ stand for functional relations (e.g., see Fig. 1). By convention, $\boldsymbol{U}$ are not explicitly shown; a bi-directed arrow between $V_i$ and $V_j$ indicates the presence of an unobserved confounder (UC) $U_k$ affecting both $V_i, V_j$, i.e., $V_i \leftarrow U_k \rightarrow V_j$.

A path is a sequence of edges where each pair of adjacent edges in the sequence share a node. We use d-separation and blocking interchangeably, following the convention in [16]. A path from a node $X$ to a node $\hat{Y}$ consists exclusively of direct arrows pointing away from $X$ is called causal; all the other non-causal paths are called spurious. The causal paths could be further categorized into the *direct* path $X \rightarrow \hat{Y}$ and the *indirect* paths, e.g., $X \rightarrow W \rightarrow \hat{Y}$ of Fig. 2(a). Let $(X \rightarrow \hat{Y})_G$, $(X \xrightarrow{i} \hat{Y})_G$ and $(X \xleftrightarrow{s} \hat{Y})_G$ denote, respectively, the direct, indirect and spurious paths between $X$ and $\hat{Y}$ in a DAG $G$. A descendant of $X$ is any node which $X$ has a causal path to (including $X$ itself). The descendant set of a set $\boldsymbol{X}$ is all descendants of any node in $\boldsymbol{X}$, which we denote by $De(\boldsymbol{X})_G$.

An intervention on a set of variables $X \subseteq \boldsymbol{V}$, denoted by $do(x)$, is an operation where values of $X$ are set to constants $x$, regardless of how they were ordinarily determined (through the functions $f_X$). We denote by $\langle M_x, P(\boldsymbol{u})\rangle$ a sub-model of a SCM $\langle M, P(u)\rangle$ induced by $do(x)$. The potential response of $\hat{Y}$ to intervention $do(x)$, denoted by $\hat{Y}_x(\boldsymbol{u})$, is the solution of $\hat{Y}$ with $\boldsymbol{U} = \boldsymbol{u}$ in the sub-model $M_x$; it can be read as the counterfactual sentence "the value that $\hat{Y}$ would have obtained in situation $\boldsymbol{U} = \boldsymbol{u}$, had $X$ been $x$." Statistically, averaging $\boldsymbol{U}$'s distribution ($P(\boldsymbol{u})$) leads to the counterfactual variable $\hat{Y}_x$. For a more detailed discussion on SCMs, please refer to [16, 2].

## 3 Counterfactual Analysis of Unequalized Classification Errors

In this section, we investigate the unequalized odds of misclassification observed in COMPAS by devising three simple thought experiments. These experiments could be generalized into a set of novel counterfactual measures, providing a fine-grained explanation of how the ER disparity of a classifier $f(\hat{\boldsymbol{pa}})$ is brought about. Throughout our analysis, we will let $X$ be the protected attribute, $\hat{Y}$ be the prediction and $Y$ be the true outcome; $\hat{\boldsymbol{PA}}$ is a set of (possible) input features of the predictor $\hat{Y}$. We will denote by value $x_1$ the disadvantaged group and $x_0$ the advantaged group. Given the space constraints, all proofs are included in the full technical report [27, Appendix A].

We consider first the impact of the direct discrimination (i.e., the direct path $X \rightarrow \hat{Y}$) on the ER disparity observed in the COMPAS. We will devise a thought experiment concerning with a Caucasian defendant who does not recidivate (i.e., $x_0, y$). Imagine a hypothetical situation where this defendant were a non-recidivating African-American ($x_1, y$), while keeping the prior convictions $W$ and other demographic information $Z$ fixed at the level that the defendant $x_0, y$ currently has. We then measure the prediction $\hat{Y}$ in this imagined world (counterfactually), compared to what the defendant currently receives from COMPAS (factually). If the prediction were different in these two situations, e.g., $\hat{Y}$

changes from 0 to 1, we could then say the path $X \rightarrow \hat{Y}$ is active, i.e., the direct discrimination against African-American defendants exists.

Figs. 3(a-b) represent this thought experiment graphically. Fig. 3(b) shows the conditional SCM $\langle M, P(\boldsymbol{u}|x_0, y)\rangle$ of the non-recidivating Caucasian defendant $(x_0, y)$: variables $X, Z, W$ are correlated by conditioning on the collider $Y$ [16, pp. 339]; we omit the true outcome $Y$ for simplicity. Using this model as the baseline (i.e.,

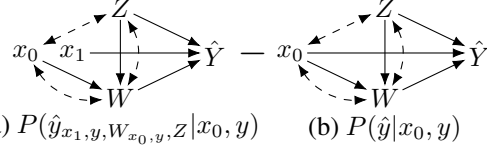

(a) $P(\hat{y}_{x_1, y, W_{x_0, y}, Z}|x_0, y)$     (b) $P(\hat{y}|x_0, y)$

Figure 3: Graphical representation of the counterfactual direct ER in COMPAS.

what factually happened in reality), we change in Fig. 3(a) the input of $X$ to the direct path $X \rightarrow \hat{Y}$ to $x_1$ (edges in $G$ represent functional relations), while keeping the value of $X$ to other variables $(W, Z)$ fixed at the baseline level $x_0, y$. In this reality, variable $Z_{x_0, y} = Z$ since $Z$ is a non-descendant node of $X$ and $Y$ [16, pp. 232]; the intervention on $Y$ is omitted since $Y$ does not directly affect the prediction $\hat{Y}$. Since the direct path $X \rightarrow \hat{Y}$ is the only difference between models of Figs. 3(a-b), the change in $\hat{Y}$ thus measure the influence of $X \rightarrow \hat{Y}$. Indeed, this hypothetical procedure could be generalized, applicable to any classifier in an arbitrary SCM, which we summarize as follows.

**Definition 1** (Counterfactual Direct Error Rate). Given a SCM $\langle M, P(\boldsymbol{u})\rangle$ and a classifier $f(\hat{\boldsymbol{pa}})$, the counterfactual direct error rate for a sub-population $x, y$ (with prediction $\hat{y} \neq y$) is defined as:

$$ER^d_{x_0, x_1}(\hat{y}|x, y) = P(\hat{y}_{x_1, y, (\hat{\boldsymbol{PA}} \backslash X)_{x_0, y}}|x, y) - P(\hat{y}_{x_0, y}|x, y) \tag{1}$$

In Eq. 1, $\hat{Y}_{x_1, y, (\hat{\boldsymbol{PA}} \backslash X)_{x_0, y}}$ could be further simplified as $\hat{Y}_{x_1, (\hat{\boldsymbol{PA}} \backslash X)_{x_0, y}}$ since $Y$ is not an input of $f(\hat{\boldsymbol{pa}})$. The subscript $(\hat{\boldsymbol{PA}} \backslash X)_{x_0, y}$ is the solution of the input features (besides $X$) $(\hat{\boldsymbol{PA}} \backslash X)(\boldsymbol{u})$ in the sub-model $M_{x_0, y}$; values of $\boldsymbol{U}$ are drawn from the distribution $P(\boldsymbol{u})$ such that $X(\boldsymbol{u}) = x, Y(\boldsymbol{u}) = y$. The query of Eq. 1 could be read as: "For an individual with the protected attribute $X = x$ and the true outcome $Y = y$, how would the prediction $\hat{Y}$ change had $X$ been $x_1$, while keeping all the other features $\hat{\boldsymbol{PA}} \backslash X$ at the level that they would attain had $X = x_0$ and $Y = y$, compared to the prediction $\hat{Y}$ she/he would receive had $X$ been $x_0$ and $Y$ been $y$?"

Similarly, we could devise a thought experiment to measure the effect of the indirect discrimination, mediated by the prior convictions $W$, i.e., the indirect path $X \rightarrow W \rightarrow \hat{Y}$. Consider again the non-recidivating Caucasian defendant $x_0, y$. We conceive a scenario where the prior convictions $W$ of the defendant $x_0, y$ changes to the level that

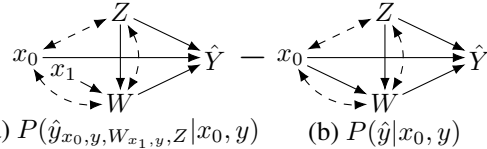

(a) $P(\hat{y}_{x_0, y, W_{x_1, y}, Z}|x_0, y)$     (b) $P(\hat{y}|x_0, y)$

Figure 4: Graphical representations of the counterfactual indirect ER in COMPAS.

it would have achieved had the defendant been a non-recidivating African-American $x_1, y$, while keeping the other features $X, Z$ fixed at the level that they currently are. Fig. 4(a) describes this hypothetical scenario: we change only input value of edge $X \rightarrow W$ to $x_1$, while keeping all the other paths untouched (at the baseline). We then measure the prediction $\hat{Y}$ in both the counterfactual (Fig. 4(a)) and factual (Fig. 4(b)) world and compare their differences. The change in the prediction of these models thus represent the influence of indirect path $X \rightarrow W \rightarrow \hat{Y}$. We generalize this thought experiment and provide an estimand of the indirect paths for any SCM and classifier $f$, namely:

**Definition 2** (Counterfactual Indirect Error Rate). Given a SCM $\langle M, P(\boldsymbol{u})\rangle$ and a classifier $f(\hat{\boldsymbol{pa}})$, the counterfactual indirect error rate for a sub-population $x, y$ (with prediction $\hat{y} \neq y$) is defined as:

$$ER^i_{x_0, x_1}(\hat{y}|x, y) = P(\hat{y}_{x_0, y, (\hat{\boldsymbol{PA}} \backslash X)_{x_1, y}}|x, y) - P(\hat{y}_{x_0, y}|x, y). \tag{2}$$

Finally, we introduce a hypothetical procedure measuring the influence of the spurious relations between the protected attribute $X$ and prediction $\hat{Y}$ through the population attributes that are non-descendants of both $X$ and $\hat{Y}$, e.g., the path $X \leftrightarrow Z \rightarrow \hat{Y}$ in Fig. 2(a). We consider a Caucasian $x_0, y$ and an African-American $x_1, y$ defendants who both would not recidivate. We measure the prediction $\hat{Y}$ these defendants would receive had they both been

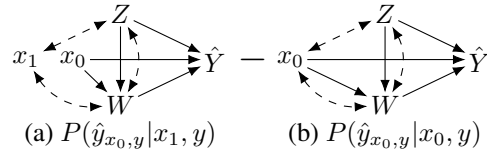

(a) $P(\hat{y}_{x_0, y}|x_1, y)$     (b) $P(\hat{y}_{x_0, y}|x_0, y)$

Figure 5: Graphical representations of the counterfactual spurious ER in COMPAS.

non-recidivating Caucasians $(x_0, y)$. Figs. 5 (a-b) describes this experimental setup. Since the causal influence of $X$ (on $\hat{Y}$) are fixed at $x_0$ in both models, the difference in $\hat{Y}$ must be due to the population characteristics that are not affected by $X$ i.e., the spurious $X - \hat{Y}$ relationships.

**Definition 3** (Counterfactual Spurious Error Rate). Given a SCM $\langle M, P(\boldsymbol{u}) \rangle$ and a classifier $f(\hat{\boldsymbol{pa}})$, the counterfactual spurious error rate for a sub-population $x, y$ (with prediction $\hat{y} \neq y$) is defined as:

$$ER^s_{x_0,x_1}(\hat{y}|y) = P(\hat{y}_{x_0,y}|x_1,y) - P(\hat{y}_{x_0,y}|x_0,y) \tag{3}$$

Def. 3 generalizes the thought experiment described above to an arbitrary SCM. In the above equation, the distribution $P(\hat{y}_{x_0,y}|x_0,y)$ coincides with $P(\hat{y}|x_0,y)$ since variable $\hat{Y}_{x_0,y} = \hat{Y}$ given that $X = x_0, Y = y$ (the composition axiom [16, Ch. 7.3]). Eq. 3 can be read as the counterfactual sentence: "For two demographics $x_0, x_1$ with the same true outcome $Y = y$, how would the prediction $\hat{Y}$ differ had they both been $x_0, y$?"

### 3.1 Properties of Counterfactual Error Rates

**Theorem 1.** *Given a SCM $\langle M, P(\boldsymbol{u}) \rangle$ and a classifier $f(\hat{\boldsymbol{pa}})$, for any $x_0, x_1, x, \hat{y}, y$, the counterfactual ERs of Defs. 1-3 obey the following properties : (1) $(X \nrightarrow Y)_{G_{|Y}} \Rightarrow ER^d_{x_0,x_1}(\hat{y}|x,y) = 0$; (2) $|(X \xrightarrow{i} Y)_{G_{|Y}}| = 0 \Rightarrow ER^i_{x_0,x_1}(\hat{y}|x,y) = 0$; (3) $|(X \xleftrightarrow{s} Y)_{G_{|Y}}| = 0 \Rightarrow ER^s_{x_0,x_1}(\hat{y}|x,y) = 0$, where $G_{|Y}$ is the causal diagram of a conditional SCM $\langle M_y, P(\boldsymbol{u}|y) \rangle$.*

The conditional causal diagram $G_{|Y}$ is obtained from the original model $G$ by (1) removing the node $Y$ and (2) adding bi-directed arrows between nodes whose associated exogenous variables are correlated in $P(\boldsymbol{u}|y)$[1] (e.g., Fig. 3(b)). Thm. 1 says that Defs. 1-3 provide *prima facie* evidence for discrimination detection. For instance, $ER^d_{x_0,x_1}(\hat{y}|x,y) \neq 0$ implies that the path $X \to \hat{Y}$ is active, i.e., the direct discrimination exists. It is expected that the proposed counterfactual measures capture the relative strength of different active pathways connecting node $X$ and $\hat{Y}$ in the underlying SCM. We now derive how the counterfactual ERs are quantitatively related with the unequalized odds of misclassification induced by an arbitrary classifier.

**Theorem 2** (Causal Explanation Formula of Equalized Odds). *For any $x_0, x_1, \hat{y}, y$, $ER_{x_0,x_1}(\hat{y}|x,y)$, $ER^d_{x_0,x_1}(\hat{y}|x,y)$, $ER^i_{x_0,x_1}(\hat{y}|x,y)$ and $ER^s_{x_0,x_1}(\hat{y}|y)$ obey the following non-parametric relationship:*

$$ER_{x_0,x_1}(\hat{y}|y) = ER^d_{x_0,x_1}(\hat{y}|x_0,y) - ER^i_{x_1,x_0}(\hat{y}|x_0,y) - ER^s_{x_1,x_0}(\hat{y}|y). \tag{4}$$

Thm. 2 guarantees that the disparate ER with the transition from $x_0$ to $x_1$ is equal to the sum of the counterfactual direct ER with this transition, *minus* the indirect and spurious ER with *reverse* transition, from $x_1$ to $x_0$, on the sub-population $x_0, y$. Together with Thm. 1, each decomposing term in Eq. 4 thus estimates the adverse impact of its corresponding discriminatory mechanism on the total ER disparity. For instance, in COMPAS, $ER^d_{x_0,x_1}(\hat{y}_1|x_0,y)$ explains how much the direct racial discrimination accounts for the unequalized false positive rate $ER_{x_0,x_1}(\hat{y}_1|y_0)$ between non-recidivating African American $(x_1, y)$ and Caucasian $(x_0, y)$ defendants. Perhaps surprisingly, this result holds non-parametrically, which means that the counterfactual ERs decompose following Thm. 2 for any functional form of the classifier and the underlying causal models where the dataset was generated. Owed to their generality and ubiquity, we refer to this equation as the "Causal Explanation Formula" for the disparate ER in classification tasks.

**Connections with Other Counterfactual Measures** Defs. 1-3 can be seen as a generalization of the marginal counterfactual measures, including the counterfactual effects introduced in [28] and the natural effects in [17, 11, 15]. Unable to consider the additional evidence (in classification, the true outcome $Y = y$), the fairness analysis framework based on these marginal measures fails to provide a fine-grained quantitative explanation of the ER disparity (as in, Thm. 2). The counterfactual fairness [10] is another counterfactual measure. As noted in [28], however, it considers only the effects along the causal paths from the protected attribute $X$ and the outcome $\hat{Y}$, thus unable to provide a full account of the $X - \hat{Y}$ associations, including the spurious relations. We provide in Appendix B [27] a more detailed discussion about the relationships between our measures and the existing ones.

# 4 Estimating Counterfactual Error Rates

The Explanation Formula provides the precise relation between the counterfactual ERs, but it does not specify how they should be estimated from data. When the underlying SCM is provided, the counterfactual direct, indirect and spurious ERs (Defs. 1-3) are all well-defined and computable via the three-step algorithm of "predictions, interventions and counterfactuals" described in [16, Ch. 7.1].

However, the SCMs are not fully known in many applications, and one must estimate the proposed counterfactual measures from the passively-collected (observational) data. Let a classifier $f(\hat{pa})$ be denoted by $f(\hat{w}, \hat{z})$, where $\hat{Z} \subseteq \hat{PA}$ are non-descendants of both $X$ and $Y$ and the subset of features $\hat{W} = \hat{PA} \backslash \hat{Z}$. We first characterize a set of classifiers where such estimation is still feasible.

**Definition 4** (Explanation Criterion). Given a DAG $G$ and a classifier $\hat{y} \leftarrow f(\hat{w}, \hat{z})$, a set of covariates $C$ satisfies the *explanation criterion* relative to $f$ (called the explaining set) if and only if (1) $\hat{Z} \subseteq C$; (2) $C \cap Forb(\{X, Y\}, \hat{W} \backslash X) = \emptyset$ where $Forb(\{X, Y\}, \hat{W} \backslash X)$ is a set of descendants $W_i \in De(W)_G$ for some $W \notin \{X, Y\}$ on a proper causal path[2] from $\{X, Y\}$ to $\hat{W} \backslash X$ in $G$; and (3) all spurious paths from $\{X, Y\}$ to $\hat{W} \backslash X$ in $G$ are blocked by $C$. A classifier $f$ is *counterfactually explainable* (ctf-explainable) if and only if it has an explaining set $C$ satisfying Conditions 1-3.

Consider again the COMPAS model of Fig. 1. The classifier $f(x, w, z)$ has input features $\hat{W} = \{X, W\}$ and $\hat{Z} = \{Z\}$. The set $C = \{Z\}$ does not satisfy the explanation criterion relative to $f$ since it does not block the spurious path $Y \leftarrow W$. Indeed, one could show that there exists no set $C$ satisfying Def. 4 relative to $f$, i.e., $f(x, w, z)$ is not ctf-explainable. However, if we remove the prior convictions $W$ from the feature set, the new classifier $f(x, z)$ is ctf-explainable with $C = \{Z\}$: $\hat{Z} = C = \{Z\}$ satisfies Condition 1; Conditions 2-3 follow immediately since $\hat{W} \backslash X = \emptyset$.

Defs. 4 constitutes a sufficient condition upon which the counterfactual ERs could, at least in principle, be estimated from the observational data. This yields identification formulas as shown next:

**Theorem 3.** *Given a causal diagram $G$ and a classifier $f(\hat{w}, \hat{z})$, if $f$ is ctf-explainable (Def. 4) with an explaining set $C$, $ER^d_{x_0,x_1}(\hat{y}|x, y)$, $ER^i_{x_0,x_1}(\hat{y}|x, y)$ and $ER^s_{x_0,x_1}(\hat{y}|y)$ can be estimated as follows:*

$$ER^d_{x_0,x_1}(\hat{y}|x, y) = \sum_{\hat{w},c}(P(\hat{y}_{x_1,\hat{w}\backslash x,\hat{z}}) - P(\hat{y}_{x_0,\hat{w}\backslash x,\hat{z}}))P(\hat{w}\backslash x|x_0, c, y)P(c|x, y), \quad (5)$$

$$ER^i_{x_0,x_1}(\hat{y}|x, y) = \sum_{\hat{w},c}P(\hat{y}_{x_1,\hat{w}\backslash x,\hat{z}})(P(\hat{w}\backslash x|x_1, c, y) - P(\hat{w}\backslash x|x_0, c, y))P(c|x, y), \quad (6)$$

$$ER^s_{x_0,x_1}(\hat{y}|y) = \sum_{\hat{w},c}P(\hat{y}_{x_1,\hat{w}\backslash x,\hat{z}})P(\hat{w}\backslash x|x_1, c, y)(P(c|x_1, y) - P(c|x_0, y)). \quad (7)$$

*where $P(\hat{y}_{\hat{w},\hat{z}})$ is well-defined, computable from the classifier $f(\hat{w}, \hat{z})^3$.*

In Eqs. 5-7, the conditional distributions $P(c|x, y)$ and $P(\hat{w}\backslash x|x_0, c, y)$ do not involve any counterfactual variable, which means that they are readily estimable by any method from the observational data (e.g., through deep nets). Continuing from the COMPAS example, we could thus estimate the counterfactual ERs of $f(x, z)$ from the distribution $P(x, y, z, w)$ using Thm. 3 with $C = \{Z\}$.

**Inverse Propensity Weighting Estimators**  Eqs. 5-7 involve summing over all possible values of $\hat{W}, C$, which may present computational and sample complexity challenges as the cardinalities of $\hat{W}, C$ grow very rapidly. There exist robust statistical estimation techniques, known as the inverse propensity weighting (IPW) [12, 18], to circumvent such issues. Given the observed data $\mathcal{D} = \{Y_i, \hat{W}_i, C_i\}_{i=1}^n$, we propose the IPW estimator for $ER^d_{x_0,x_1}(\hat{y}|x, y)$ as follows:

$$\hat{ER}^d_{x_0,x_1}(\hat{y}|x, y) = \frac{1}{n}\sum_{i=1}^n(P(\hat{y}_{x_1,\hat{W}_i\backslash X_i,\hat{Z}_i}) - P(\hat{y}_{x_0,\hat{W}_i\backslash X_i,\hat{Z}_i}))\frac{\hat{P}(x|C_i, y)I_{\{X_i=x_0,Y_i=y\}}}{\hat{P}(x_0|C_i, y)\hat{P}(x, y)}, \quad (8)$$

where $I_{\{\cdot\}}$ is an indicator function and $\hat{P}(x, y)$ is the sample mean estimator of $P(x, y)$ ($X, Y$ are finite). $\hat{P}(x|c, y)$ is a reliable estimator of the conditional distributions $P(x|c, y)$ and, in practice, could be estimated by assuming some parametric models such as logistic regression.

| **Algorithm 1:** FindExpSet | **Algorithm 2:** Causal-SFFS |
|---|---|
| **Input:** Feature set $\{\hat{\boldsymbol{W}}, \hat{\boldsymbol{Z}}\}$, DAG $G = \langle \boldsymbol{V}, \boldsymbol{E} \rangle$ <br> **Output:** Explaining set $\boldsymbol{C}$ (Def. 4) relative to $f(\hat{\boldsymbol{w}}, \hat{\boldsymbol{z}})$ in $G$, or $\perp$ if $f$ is not ctf-explainable. <br> 1: Apply *FindSep* [22] to find a set $\boldsymbol{C}$ with $\hat{\boldsymbol{Z}} \subseteq \boldsymbol{C} \subseteq \boldsymbol{V} \backslash Forb(\{X,Y\}, \hat{\boldsymbol{W}} \backslash X)$ such that it d-separates $\{X,Y\}$ and $\hat{\boldsymbol{W}} \backslash X$ in $G^{pbd}_{\{X,Y\}, \hat{\boldsymbol{W}} \backslash X}$. <br> 2: **return** $\boldsymbol{C}$ | **Input:** Samples $\boldsymbol{\mathcal{D}} = \{Y_i, \boldsymbol{V}_i\}_{i=1}^n$, a causal diagram $G$ <br> **Output:** A family of ctf-explainable classifiers $\boldsymbol{\mathcal{F}}$ <br> **Initialization:** $\hat{\boldsymbol{PA}}_0 = \emptyset$, $k = 0$. <br> 1: **while** $k < |\boldsymbol{V}|$ **do** <br> 2:     Let subset $\hat{\boldsymbol{V}}_k$ be defined as <br>     $\{v_i \in \boldsymbol{V} \backslash \hat{\boldsymbol{PA}}_k : FindExpSet(\hat{\boldsymbol{PA}}_k \cup v_i, G) \neq \perp\}$. |

| **Algorithm 3:** Ctf-FairLearning |
|---|
| **Input:** Samples $\boldsymbol{\mathcal{D}}$, DAG $G$, $\epsilon_d, \epsilon_i, \epsilon_s > 0$ <br> **Output:** A fair classifier $f$ <br> 1: Let $\boldsymbol{\mathcal{F}} = C\text{-}SFFS(\boldsymbol{\mathcal{D}}, G)$. <br> 2: Obtain a fair classifier $f$ from $\boldsymbol{\mathcal{F}}$ by solving Eq. 9 subject to $|ER^d| \leq \epsilon_d$, $|ER^i| \leq \epsilon_i$, $|ER^s| \leq \epsilon_s$. |

3:     Let $v_{k+1} = \arg \max_{v_i \in \hat{\boldsymbol{V}}_k} J(\hat{\boldsymbol{PA}}_k \cup \{v_i\})$.
4:     Let $\hat{\boldsymbol{PA}}_{k+1} = \hat{\boldsymbol{PA}}_k \cup v_{k+1}$; $k = k + 1$.
5:     Continue with the conditional exclusion of [19, Step 2-3] and update the counter $k$.
6: **end while**
7: **return** $\boldsymbol{\mathcal{F}} = \{\forall f : \hat{\boldsymbol{PA}}_k \to \hat{Y}\}$.

**Theorem 4.** *For a ctf-explainable classifier $f(\hat{\boldsymbol{w}}, \hat{\boldsymbol{z}})$, $\hat{ER}^d_{x_0, x_1}(\hat{y}|x, y)$ (Eq. 8) is a consistent estimator for $ER^d_{x_0, x_1}(\hat{y}|x, y)$ (Eq. 5) if the model for $P(x|\boldsymbol{c}, y)$ is correctly specified.*

We provide IPW estimators for counterfactual indirect and spurious ERs in Appendix A [27].

### 4.1 Finding Adjustment Set for Explainable Classifiers

A few natural questions arise here is (1) how to systematically test whether a classifier $f$ is ctf-explainable, and (2) if so, to find a set $\boldsymbol{C}$ satisfying the explanation criterion so that the counterfactual ERs could be identified. In this section, we will develop an efficient method to answer these questions.

Given a DAG $G$, by $G^{pbd}_{\{X,Y\}, \hat{\boldsymbol{W}} \backslash X}$ we denote the proper backdoor graph obtained from $G$ by removing the first edge of every proper causal path from $\{X,Y\}$ to $\hat{\boldsymbol{W}} \backslash X$ [22]. We formulate next in graphical terms a set of identification conditions equivalent to the explanation criterion defined in Def. 4.

**Definition 5** (Constructive Explanation Criterion). Given a DAG $G$ and a classifier $f(\hat{\boldsymbol{w}}, \hat{\boldsymbol{z}})$, covariates $\boldsymbol{C}$ satisfy the *constructive explanation criterion* relative to $f$ if and only if (1) $\hat{\boldsymbol{Z}} \subseteq \boldsymbol{C} \subseteq \boldsymbol{V} \backslash Forb(\{X,Y\}, \hat{\boldsymbol{W}} \backslash X)$, where $Forb(\{X,Y\}, \hat{\boldsymbol{W}} \backslash X)$ is a set of nodes forbidden by Def. 4; (2) $\boldsymbol{C}$ d-separates $\{X,Y\}$ and $\hat{\boldsymbol{W}} \backslash X$ in the proper backdoor graph $G^{pbd}_{\{X,Y\}, \hat{\boldsymbol{W}} \backslash X}$.

**Theorem 5.** *Given a causal diagram $G$ and a classifier $f$, covariates $\boldsymbol{C}$ satisfies the explanation criterion (Def. 4) to $f$ if and only if it satisfies the constructive explanation criterion (Def. 5) to $f$.*

Thm. 5 allows us to use the algorithmic framework developed by [22] for constructing d-separating sets in DAGs. We summarize this procedure as *FindExpSet*, in Alg. 1. Specifically, the sub-routine *FindSep* find a covariates set $\boldsymbol{C}$ with $\hat{\boldsymbol{Z}} \subseteq \boldsymbol{C} \subseteq \boldsymbol{V} \backslash Forb(\{X,Y\}, \hat{\boldsymbol{W}} \backslash X)$, such that $\boldsymbol{C}$ d-separates all paths between $\{X,Y\}$ and $\hat{\boldsymbol{W}} \backslash X$ in $G^{pbd}_{\{X,Y\}, \hat{\boldsymbol{W}} \backslash X}$, i.e., the explaining set relative to classifier $f(\hat{\boldsymbol{w}}, \hat{\boldsymbol{z}})$ (Def. 4). This algorithm can be solved in $\mathcal{O}(n + m)$ runtime where $n$ is the number of nodes and $m$ is the number of edges in the proper backdoor graph $G^{pbd}_{\{X,Y\}, \hat{\boldsymbol{W}} \backslash X}$.

## 5 Achieving Equalized Counterfactual Error Rates

So far we have focused on analyzing the unequalized counterfactual ERs of an existing predictor in the environment. A more interesting problem is how to obtain an optimal classifier such that its induced counterfactual ERs along with a specific discriminatory mechanism are equalized.

Given finite samples $\boldsymbol{\mathcal{D}} = \{Y_i, \boldsymbol{V}_i\}_{i=1}^n$ drawn from $P(y, \boldsymbol{v})$ (where the protected attribute $X \in \boldsymbol{V}$), the associated causal diagram $G$, and a set of candidate ctf-explainable classifiers $\boldsymbol{\mathcal{F}}$, the goal of the supervised learning is to obtain an optimal classifier $f^*(\hat{\boldsymbol{pa}})$ from $\boldsymbol{\mathcal{F}}$ such that a loss function $L(\boldsymbol{\mathcal{D}}, f)$ measuring the distance between the prediction $\hat{Y}$ and the true outcome $Y$ is minimized. We will elaborate later about how to construct the ctf-explainable set $\boldsymbol{\mathcal{F}}$. Among the quantities evolved by Thm. 3, the counterfactual distribution $P(\hat{y}_{x, \hat{\boldsymbol{w}} \backslash x, \hat{\boldsymbol{z}}})$ is defined from the classifier $f$ and the other conditional distributions (e.g., $P(\boldsymbol{c}|x, y)$) are estimable from the data $\boldsymbol{\mathcal{D}}$. We could thus represent a counterfactual ER (e.g., direct) of a classifier $f \in \boldsymbol{\mathcal{F}}$ as a function $g(\boldsymbol{\mathcal{D}}, f)$ (e.g., Eq. 8). A fair

classifier is obtained by minimizing $L(\mathcal{D}, f)$ subject to a box constraint over $g(\mathcal{D}, f)$, namely,

$$\min_{f \in \mathcal{F}} L(\mathcal{D}, f) \ \text{ s.t. } |g(\mathcal{D}, f)| \leq \epsilon, \tag{9}$$

where $\epsilon \in \mathbb{R}^+$ and the smaller $\epsilon$ is, the fairer the learned classifier would be. In general, the constraints $|g(\mathcal{D}, f)| \leq \epsilon$ are non-convex and solving the problem of Eq. 9 seems to be difficult. However, this optimization problem could be significantly simpler in certain cases, solvable using standard convex optimization methods [3]. We provide two canonical settings that fit this requirement.

First, we assume that the features $V$ are discrete, and let $\theta_{\hat{y}, x, \hat{w} \backslash x, \hat{z}}$ denote the probabilities $P(\hat{y}_{x, \hat{w} \backslash x, \hat{z}})$. The counterfactual constraints $|g(\mathcal{D}, f)| \leq \epsilon$ are thus reducible to a set of linear inequalities on the parameter space $\{\theta\}$. Second, consider a classifier making decision based on a decision boundary $\tilde{Y} = \theta^{\mathsf{T}} \phi(x, \hat{w} \backslash x, \hat{z})$ (e.g., logistic regression), where $\phi(\cdot)$ is the basis function. The boundary $\tilde{Y}$ acts as a proxy to the prediction $\hat{Y}$. For instance, the condition $ER^d_{x_0, x_1}(\tilde{y}|x, y) = 0$ implies $ER^d_{x_0, x_1}(\hat{y}|x, y) = 0$. The same reasoning applies to the counterfactual indirect and spurious ERs. We will employ the techniques in [25] and approximate the constraints $|g(\mathcal{D}, f)| \leq \epsilon$ using the counterfactual ERs of $X$ on the boundary $\tilde{Y}$. Assume that we are interested in the mean effect and replace the quantities $P(\hat{y}_{x, \hat{w} \backslash x, \hat{z}})$ in Thm. 3 with $\theta^{\mathsf{T}} \phi(x, \hat{w} \backslash x, \hat{z})$. Given the convexity of $L(\mathcal{D}, f)$, Eq. 9 is a convex optimization problem and can thus be efficiently solved using standard methods.

## 5.1 Constructing Counterfactually Explainable Classifiers

The counterfactual explainability (Def. 4) of a classifier $f$ relies on its input feature $\hat{PA}$: the smaller the set $\hat{PA}$ is, the easier it would be to find a explaining set $C$ relative to $f(\hat{pa})$. In practice, some features contain critical information about the prediction task, which means that their exclusion could lead to poorer performance. This observation suggests a novel feature selection problem in the fairness-aware classification task: we would like to find a subset $\hat{PA}$ from the available features $V$ such that each classifier in the candidate set $\mathcal{F} = \{\forall f : \hat{PA} \to \hat{Y}\}$ is ctf-explainable, without significant loss of prediction accuracy.

Our solution builds on the procedure *FindExpSet* (Alg. 1) and the classic method of Sequential Floating Forward Selection (*SFFS*) [19]. Let $\hat{PA}_k$ be the set of $k$ features. The score function $J(\hat{pa}_k)$ evaluates the candidate subset $\hat{PA}_k$ and returns a measure of its "goodness". In practice, this score could be obtained by computing the statistical measures of dependence, or by evaluating the best in-class predictive accuracy for classifiers in $\{\forall f : \hat{PA}_k \to \hat{Y}\}$ on the validation data. We denote our method by Causal *SFFS* (*C-SFFS*) and summarize it in Alg. 2. Starting with a subset $\hat{PA}_k$, *C-SFFS* (Step 2-3) adds one feature which gives the highest score $J$. *FindExpSet* ensures that the resulting subset $\hat{PA}_{k+1}$ induces a ctf-explainable classifier $f(\hat{pa}_{k+1})$. Step 5 repeatedly removes the least significant feature $v_d$ from the newly-formed $\hat{PA}_k$ until no feature could be excluded to improve the score $J$. During the exclusion phase, we do not apply *FindExpSet*, since removing features from a ctf-explainable classifier does not violate the explanation criterion (Def. 4). It follows immediately from the soundness of *FindExpSet* that *C-SFFS* always returns a ctf-explainable set $\mathcal{F}$.

**Theorem 6.** *For $\mathcal{F} = C\text{-}SFFS(\mathcal{D}, G)$, each classifier $f \in \mathcal{F}$ is ctf-explainable.*

We summarize in Alg. 3 the procedure of training an optimal classifier satisfying the fairness constraints over the counterfactual ERs. $ER^d$, $ER^i$, and $ER^s$ stand for the counterfactual quantities $ER^d_{x_0, x_1}(\hat{y}|x_0, y)$, $ER^i_{x_1, x_0}(\hat{y}|x_0, y)$, and $ER^s_{x_1, x_0}(\hat{y}|y)$, respectively. We use *C-SFFS* (Alg. 2) to obtain a candidate set $\mathcal{F}$ such that each $f \in \mathcal{F}$ is ctf-explainable. The fair classifier is computed by solving the optimization problem in Eq. 9 subject to the box constraints over $ER^d$, $ER^i$, and $ER^s$.

## 6 Simulations and Experiments

In this section, we will illustrate our approach on both synthetic and real datasets. We focus on the *false positive rate* $ER_{x_0, x_1}(\hat{y}_1|y_0)$ across demographics $x_0 = 0, x_1 = 1$, where $\hat{y}_1 = 1, y_0 = 0$, and the corresponding components $ER^d_{x_0, x_1}(\hat{y}_1|x_0, y_0)$, $ER^i_{x_1, x_0}(\hat{y}_1|x_0, y_0)$ and $ER^s_{x_1, x_0}(\hat{y}_1|y_0)$ (following Thm. 2). We shorten the notation and write $ER_{x_0, x_1}(\hat{y}_1|y_0) = ER$, and similarly to $ER^d$, $ER^i$ and $ER^s$. Details of the experiments are provided in Appendix C [27].

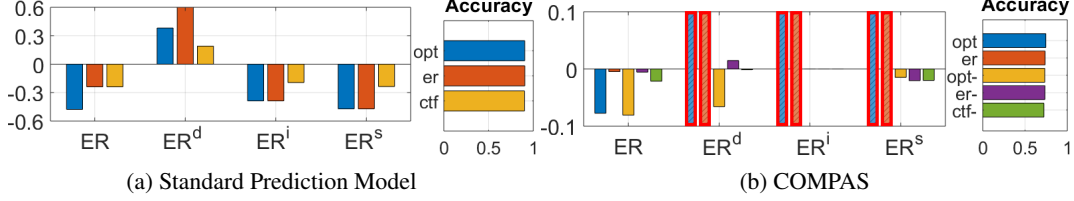

(a) Standard Prediction Model                          (b) COMPAS

Figure 7: Results of Experiments 1-2. Measures that are not estimable via the explanation criterion are shaded and highlighted. *ER* stands for the false positive rate $ER_{x_0,x_1}(\hat{y}_1|y_0)$; $ER^d$, $ER^i$ and $ER^s$ represent the corresponding counterfactual direct, indirect, and spurious ERs (Thm. 2). Classifier $f_{opt}$, $f_{er}$, and $f_{ctf}$ in Exp. 1 correspond to, respectively, color blue, orange, and yellow in Fig. (a); $f_{opt}$, $f_{er}$, $f_{opt\text{-}}$, $f_{er\text{-}}$, and $f_{ctf\text{-}}$ in Exp. 2 correspond to blue, orange, yellow, purple, and green in Fig. (b).

**Experiment 1: Standard Prediction Model**  We consider a generalized COMPAS model containing the common descendant $D$, shown in Fig. 6, which we call here the *standard fairness prediction model* (for short, standard prediction model). We train two classifiers with the same feature set $\{X, W, Z, D\}$ where the first is obtained via the standard, unconstrained optimization, which we call $f_{opt}$, and the second one constrains the disparate *ER* to half of that of $f_{opt}$, which we call $f_{er}$. We

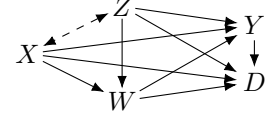

Figure 6: Standard fairness prediction model

further compute the counterfactual ERs (Defs. 1-3). The results are shown in Fig. 7(a). We first confirm that the procedure $f_{er}$ is sound in the sense that $f_{eo}$ (90.4%) achieves a comparable predictive accuracy to $f_{opt}$ (90.4%) while reducing the disparate ER in half ($ER_{er} = -0.238$, $ER_{opt} = -0.476$). Second, $ER^d$ is larger in $f_{er}$ ($ER^d_{eo} = 0.620$) than in the unconstrained $f_{opt}$ ($ER^d_{opt} = 0.381$). This materializes the concern acknowledged in [7], namely, that optimizing based on *ER* may not be enforcing any type of real-life fairness notion related to the underlying causal mechanism. To circumvent this issue, we train a classifier with the same feature set such that its counterfactual ERs are reduced to half of that of the unconstrained $f_{opt}$, called $f_{ctf}$. The results (Fig. 7(a)) support the counterfactual approach: $f_{ctf}$ (90.1%) reports *ER* comparable to $f_{er}$ ($ER_{ctf} = -0.238$), but a smaller significant direct, indirect and spurious ER disparities ($ER^d_{ctf} = 0.191$, $ER^d_{ctf} = -0.194$, $ER^d_{ctf} = -0.236$).

**Experiment 2: COMPAS**  In the COMPAS model of Fig. 1, we are interested in predicting whether a defendant would recidivate, while avoiding the direct discrimination (the threshold $\epsilon = 0.01$). We compute a classifier $f_{er}$ with a feature set $\{X, Z, W\}$ subject to $|ER_{er}| \leq \epsilon$. We also include an unconstrained classifier $f_{opt}$ as the baseline. The results (Fig. 7(b)) reveal that $f_{er}$ (73.7%) and $f_{opt}$ (74.6%) are comparable in prediction accuracy while $f_{er}$ has much smaller disparate ER ($ER_{er} = -0.005$, $ER_{opt} = -0.077$). Given that the underlying causal model is not fully known, we could only estimate the counterfactual direct ER from passively-collected samples. Since classifiers with feature set $\{X, W, Z\}$ are not ctf-explainable in the COMPAS model (Def.4), $ER^d$ of $f_{er}$ and $f_{opt}$ cannot be identified via Thm. 3. Previous analysis (Experiment 1) implies that $ER^d$ could be significant even when *ER* is small, which suggests one should be wary of the direct discrimination of $f_{er}$ and $f_{opt}$. To overcome this issue, we remove $W$ from the feature set and obtain $f_{opt\text{-}}$ and $f_{er\text{-}}$ following a similar procedure. We estimate their $ER^d$ via Thm. 3 with covariates $C = \{Z\}$. The results show that the direct discrimination are significant in both $f_{er\text{-}}$ and $f_{opt\text{-}}$ ($ER^d_{eo} = 0.015$, $ER^d_{opt-} = -0.066$). To remove the direct discrimination, we train a classifier $f_{ctf\text{-}}$ following Alg. 3 with the features $\{X, Z\}$ and $\epsilon_d = \epsilon$. The results support the efficacy of Alg. 3: $f_{ctf\text{-}}$ performs slightly worse in prediction accuracy (72.1%) but ascertains no direct discrimination ($ER^d_{ctf-} = -0.001$).

## 7 Conclusions

We introduced a new family of counterfactual measures capable of explaining disparities in the misclassification rates (false positive and false negative) across different demographics in terms of the causal mechanisms underlying the specific prediction process. We then developed machinery based on these measures to allow data scientists (1) to diagnose whether a classifier is operating in a discriminatory fashion against specific groups, and (2) to learn a new classifier subject to fairness constraints in terms of fine-grained misclassification rates. In practice, this approach constitutes a formal solution to the notorious lack of interpretability of the equalized odds. We hope the causal machinery put forwarded here will help data scientists to analyze already deployed systems as well as to construct new classifiers that are fair even when the training data comes from an unfair world.

**Acknowledgments**

This research is supported in parts by grants from IBM Research, Adobe Research, NSF IIS-1704352, and IIS-1750807 (CAREER).

## Footnotes

[1] $G_{|Y}$ explicitly represents the change of information flow due to conditioning on the true outcome $Y$: the information via arrows pointing away from $Y$ is intercepted; measuring the collider $Y$ makes its (marginally independent) common causes dependent, also known as the "explaining away" effect [16, pp. 339].

[2]A causal path from $\{X, Y\}$ to $\hat{W}\backslash X$ is proper if it does not intersect $\{X, Y\}$ except at the end point [20].

[3]For a deterministic $f(\hat{w}, \hat{z})$, the probabilities $P(\hat{y}_{\hat{w},\hat{z}}) = I_{\{\hat{y}=f(\hat{w},\hat{z})\}}$ where $I_{\{\cdot\}}$ is an indicator function.

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
