[Reviews · NeurIPS 2018]

Reviewer 1



I acknowledge having read the rebuttal. Fairness is a complicated and an important matter. Due to the nature of the problem, there might not be a universal characterization of it, but if a criterion is proposed it should be followed by a compelling story and a reasonable explanation for why we should consider this criterion. This paper provides a new (causal) interpretation of equalized odds (EO), an associative measure that has been used as a framework to talk about discrimination in classification problems. The central point of the paper is to learn a fair classifier by constraining each of the three (causal) components of EO (i.e. direct effect, indirect effect, spurious effect) to be small, and not EO itself. There might be situations where this would make sense but authors don’t discuss it and it is unclear why we should care about it in general. Another drawback of the proposed procedure is the way new unseen instances are being handled. In a typical supervised setting, we use train data from p(y, x) to learn classifiers or outcome regression models (via a bias/variance trade off) in order to achieve a good out of sample performance. Both train and test data are drawn from the same distribution, otherwise this leads to what is called “covariate shift” in ML. In this paper, to learn a fair classifier, authors propose to minimize a loss function subject to those three fairness constraints, eq (4); this implicitly transfers the problem to some other distribution, p*(y, x) where all the constraints are satisfied (note that p* is potentially different than p). This is fine as far as that goes, but in the experiment, they are computing out of sample performance of the “fair classifier” (using data that comes from the observed distribution). It’s not quite clear why it makes sense to learn a classifier on one distribution, p*, and check its performance on a different distribution, p. There might be multiple fixes to this issue, or a valid explanation why this isn’t a problem at all. The authors do not clarify (nor they seem to understand the fundamental issue).

Reviewer 2



The aim of the paper is to decompose the quantitative measure of discrimination called the equalized odds (EO). The EO measures the extent to which gender, race or other protected characteristics are used in classification. The EO is decomposed into a direct effect, indirect effect and spurious effect. The paper also provides an algorithm for training classifiers with penalization to ensure that these effects are small. The aim is clear and useful since it is important to minimize the impact of unfairness, with individual measures of the various mechanisms. The significance of the paper is not clear since it appears to just extend the results of Zhang and Barenboim (2018) by defining causal effects which condition on Y. Clearly there are technical details that are addressed which describe estimation under the graph which conditions on Y. However the level of novelty does not obviously meet the required criteria. Additionally, the causal explanation formula is not particularly intuitive. The EO is decomposed as EO=DE-IE-SE, with some reversal of the effects. While this seems technically correct, its usefulness is unclear since it is not intuitive. Without some sort of additive decomposition, as with the decomposition of a causal effect into a direct and indirect effect, it may not be helpful in practice. A difference in effects is useful to decompose but, more generally, it may be a ratio or other contrasts that are of interest. This is a future concern as it may not be as straightforward and solving the difference case would be sufficiently interesting. The naming of the EO as odds is a bit misleading since it is not a ratio. However I understand that it is already in the literature.

Reviewer 3



This is a well-written, detailed paper on an important topic. The paper seeks to decompose sources of discrimination into spurious, direct and indirect effects. It strengthens connections between the causal literature on direct effects to the literature on discrimination in classification by connecting to the notion of Equalized Odds. Specific comments: (1) It is not clear what is the overall goal here. Other papers e.g. Nabi and Shpitser talk about trying to remove the effects of "unfair" pathways. Are there specific pathways that the authors think are "unfair"? Why does it make sense to condition on y? This is done in the EO measure, but is that a good idea? Some contrast relating conditional vs. marginal measure would be useful. (2) There is a close connection between what are called in this paper "conditional" counterfactual effects and papers on the effect of treatment on the treated. Also effects in strata defined by post-treatment outcomes. For example: Marshall M. Joffe (2002) Using information on realized effects to determine prospective causal effects. https://doi.org/10.1111/1467-9868.00311 (3) The authors in some places suggest that the model in Figure 1(b) represents a canonical graph for this problem. This is inaccurate because the SCM here presumes that there are no unobserved variables (so that the U terms are independent). In general there could be unobserved latent variables giving rise to confounding between any pair of variables in X,Z,Y,M. (hat{Y} is not confounded since it is a function of predictors that are, by construction, observed.) (4) Lines 122-124: The authors say that additional evidence (Y=y) mean that "the original causal diagram no longer captures in an explicit fashion all the underlying mechanisms that relate X and hat{Y}" This seems to be a category mistake: the presence of additional evidence can induce (probabilistic) independence or dependence but it does not change mechanisms! To address this, the authors suggest modifying the causal diagram (line 126). It is not clear why one couldn't just condition on y and use d-separation/connection as usual. (As noted, it seems a conceptual error to think of information changing a causal process.) (5) In the definition of conditional causal model why is M_e required. Note that in the subsequent definitions interventions are on x_0 and x_1, while the "evidence" is just x (which could be some other value that is not x_0 or x_1). Related: why are the potential outcomes in (1), (2), (3) subscripted by "e" (or "y"). Why does it make sense to consider intervening on e or y (since this is a response (that doesn't directly affect hat{Y}. Could this be left out as a subscript / intervention here? (Obviously e or y need to be conditioned on.) I guess the text at lines 144-145 makes the point that these could be omitted - though it seems there is an error here (see below) - but this doesn't explain why they were included in the first place! (6) Lines 233-236: The authors suggest that direct and indirect effects could be learned from a randomized experiment. Though true of "controlled direct effects", this is not true of "natural" direct and indirect effects such as (1) and (2) that involve nested counterfactuals y_{x1,.. W_{x0}}. J. Robins and T.S. Richardson. (2011). Alternative Graphical Causal Models and the Identification of Direct Effects. Chapter 6. Causality and Psychopathology: Finding the Determinants of Disorders and their Cures, pp. 1-52. http://www.csss.washington.edu/Papers/wp100.pdf As noted by Robins & Richardson, the assumption of independent errors in an SCM is not empirically testable via randomized experiments (since it implies independence of cross-world counterfactuals such as W(x0) and Y(x1,w). Minor comments: line 3 Hartz et al is not in the Bibliography. [Hardt et al??] line 43 functions => function lines 144-145. "Since Y is a non-descendant node of W, hat{Y}, it has no effect in the counterfactuals before the conditioning bar in the factors of Eq.1" this should be: Since W, hat_{Y} are non-descendants of Y... no effect *on*... it is also a bit hard to follow because Y doesn't show up in these terms, it seems implicitly e=Y here (?) line 155: defender => defendant (?) [Unless this is about football players!] line 170 Equation (3) SE_{x0,x1}. This notation is a bit inconsistent with that for DE and IE because x0 and x1 are not counterfactuals. line 184 featbure line 199 may not been line 208 remove "as" line 240 and 241. End of clause (1) and of clause (3). "blocked by by C" in which graph? line 249. where P(hat{y}_{x,w}). I think this should be: P(hat{y}_{x,w,hat{z}}) line 262 the the line 274: Theorem 3. The assumption of an SCM with independent errors is required here. (This is not empirically testable with any randomized experiment.) line 299 (before) Figure 4 (b) Experiment 2. What do the bars going from 0.1 to -0.1 mean? What is the difference between these and no bar at all? lines 304-318. Is it the case that the counterfactual conditional contrasts (DE, IE etc.,) are known here because the generating mechanism / SCM is known? (In other words, no estimation is being carried out.) line 323. smaller equalized *odds* line 330 line 328 to 330. In the context of this real data example why would one believe that the conditions in Theorem 2 apply (i.e. ctf explainability). Is it reasonable to assume there are no unmeasured confounders? ================================================== The authors reply: "3. Conditional Causal Models (R3) The conditional causal model does not modify the existing causal mechanisms, but only highlights the spurious and causal pathways that are not distinguishable by d-separation in the (standard) original causal model." This sounds reasonable, but it is *not* in keeping with the text in the original paper. Specifically, lines 122-127 in the original paper say the following: "This phenomenon exposes that, whenever an additional evidence is provided (Y = y), the original causal diagram no longer captures in an explicit fashion all the underlying mechanisms that relate X and hat{Y} We propose a simple modification of the causal diagram to account for the aforementioned extra mechanisms arising due to conditioning." My guess is that the word "mechanism" here is the source of the problems. This suggests changing the "underlying" SCM. I think that if the authors replaced this with "active pathways" it would have caused much less confusion.